# Current Status of Research on Small Extracellular Vesicles for the Diagnosis and Treatment of Urological Tumors

**DOI:** 10.3390/cancers15010100

**Published:** 2022-12-23

**Authors:** Mengting Zhang, Yukang Lu, Lanfeng Wang, Yiping Mao, Xinyi Hu, Zhiping Chen

**Affiliations:** 1The First School of Clinical Medicine, Gannan Medical University, Ganzhou 341000, China; 2Department of Laboratory Medicine, First Affiliated Hospital of Gannan Medical University, Ganzhou 341000, China; 3Department of Nephrology, First Affiliated Hospital of Gannan Medical University, Ganzhou 341000, China

**Keywords:** small extracellular vesicles, urological tumors, drug carriers, angiogenesis, tumor vaccines

## Abstract

**Simple Summary:**

Small extracellular vesicles (sEVs) play an important role in the occurrence and development of various diseases, exhibiting the characteristics of wide acquisition and strong specificity. Due to the difficulties in the diagnosis and treatment of urological tumors, this article mainly reviews the function of sEVs in urinary tumors and their related application in the diagnosis and treatment of such tumors. The contents herein can provide new directions for the early diagnosis and individualized treatment of urinary tumors.

**Abstract:**

Extracellular vesicles (EVs) are important mediators of communication between tumor cells and normal cells. These vesicles are rich in a variety of contents such as RNA, DNA, and proteins, and can be involved in angiogenesis, epithelial-mesenchymal transition, the formation of pre-metastatic ecological niches, and the regulation of the tumor microenvironment. Small extracellular vesicles (sEVs) are a type of EVs. Currently, the main treatments for urological tumors are surgery, radiotherapy, and targeted therapy. However, urological tumors are difficult to diagnose and treat due to their high metastatic rate, tendency to develop drug resistance, and the low sensitivity of liquid biopsies. Numerous studies have shown that sEVs offer novel therapeutic options for tumor treatment, such as tumor vaccines and tumor drug carriers. sEVs have attracted a great deal of attention owing to their contribution to in intercellular communication, and as novel biomarkers, and role in the treatment of urological tumors. This article reviews the research and applications of sEVs in the diagnosis and treatment of urological tumors.

## 1. Background

Urological tumors can occur in any part of the urinary system. Common malignant forms of these tumors include renal cell carcinoma (RCC), prostate cancer (PCa), and bladder cancer (BC). According to the 2019 American Cancer Society, the number of new cases of urological tumors in the United States in 2019 was as high as 158,220, and while estimated number of deaths resulting from them was 33,420 [1]. These tumors are a significant cause of death in men worldwide [2]. Currently, the main treatments for urological tumors include surgery, radiotherapy, and targeted therapy, but the monitoring and treatment of these cancers yield poor outcomes [3]. The deep location of urologic tumors within the body makes renders them difficult to be diagnosed early [4]. Therefore, there exists a need for novel biomarkers and therapeutic tools to improve the diagnostic efficacy and survival of urologic tumors. Small extracellular vesicles (sEVs) have the potential to be developed as biomarkers. All cells release sEVs, which is a feature that makes them highly heterogeneous throughout the body and different cell types [5,6]. Currently, research on the development of sEVs as diagnostic markers for urological tumors has primarily focused on the development of sEVs from serum, urine, and plasma sources, especially for sEVs carrying microRNAs. Bodily fluid-derived sEVs can be used for the early diagnosis of tumors and prognostic monitoring. In particular, sEVs harboring RNAs for PCa detection have been included in the National Comprehensive Cancer Network guidelines as biomarkers for early prostate cancer detection [7]. Due to the characteristic membrane structure of sEVs, they can effectively block RNAases and proteases. Thus, sEVs can also be used as drug delivery vehicles to deliver anti-tumor drugs or genes associated with drug resistance to target tumors, thereby reducing their resistance to drugs and assisting in tumor therapy; the development of novel tumor vaccines can also inhibit tumor growth [8,9]. This article briefly describes the existing literature on sEVs and relevance and potential application value in the diagnosis and treatment of urological malignancies.

## 2. Overview of Small Extracellular Vesicles

Extracellular vesicles (EVs) are small lipid membrane-bound vesicles with diameters between 30–2000 nm that are secreted by different cells into the extracellular space [10,11]. The MISEV 2018 guidelines update the nomenclature rules to classify EVs into small extracellular vesicles and medium/large extracellular vesicles, depending on their size [12,13,14]. Of these three types, research has primarily focused on sEVs, and this is mainly because their formation pathways have been more thoroughly studied and they are more stable in bodily fluids. Moreover, sEVs can act as bridges for cellular communication between donor and recipient cells [15]. sEVs contain a variety of contents including nucleic acids and proteins, which allows them to perform important roles in the occurrence and development of various diseases [16].

### 2.1. Composition of Small Extracellular Vesicles

sEVs are largely comprised of EVs proteins, nucleic acids, and EVs lipids [17]. They are found in a variety of bodily fluids, including blood, urine, cerebrospinal fluid, ascites, and milk, among others [18]. Their contained proteins include heat shock proteins (hsp 70, hsp 90, etc.) [19], membrane proteins (CD 63, CD81, CD9, etc.) [20], multivesicular body (MVB)-forming proteins (Alix, TSG101, etc.) [21], and cytoskeletal proteins, which are involved in the formation and release of sEVs. Their contained lipids include cholesterol [21], ceramides [22], and sphingolipids [23], which can regulate the biological activity of sEVs. Their contained nucleic acids include DNA, mRNA, microRNA, circRNA, and lncRNA [24,25]. sEVs largely contribute to the development of diseases through their mechanisms involving these cargo substances.

### 2.2. Formation of Small Extracellular Vesicles

The mechanism of small extracellular vesicle formation is still not fully understood, with the most classical pathway being that involving the endosomal sorting complex required for transport (ESRCT) [26]. The specific formation process of sEVs includes the following. First, the cell membrane inverts to form endosomes, which carry genetic material and fuse with each other to form early nuclear endosomes. Subsequently, these early endosomes form MVBs via the ESRCT pathway, which are then released into the extracellular space by traction of the Ras-associated GTP-binding proteins 27a and 27b (Rab 27a and Rab 27b) (Figure 1) [27]. Also, sEVs can form through budding from the membrane.

### 2.3. Isolation Techniques for Small Extracellular Vesicles

There are currently many methods used for the isolation of small extracellular vesicles, but there is no consensus on the best extraction method. Commonly used methods include: differential centrifugation, size exclusion, precipitation, and immunoaffinity separation [28]. Each method features advantages and disadvantages, and the most suitable separation method should be selected by analyzing the downstream experimental needs [13,29]. (i) Currently, differential ultracentrifugation is the most commonly used method for isolating and concentrating sEVs. It usually involves low-speed centrifugation to remove cells and large vesicles, followed by ultra-high-speed centrifugation to collect sEVs. (ii) Density gradient centrifugation is a more rigorous form of ultra-centrifugation, where vesicles of different densities settle at different rates on a gradient. (iii) Chemicals such as polyethylene glycol can reduce the solubility of sEVs thereby causing them to precipitate, followed by low-speed centrifugation to obtain sEVs. (iv) Immunocapture involves the separation of vesicles coated with magnetic beads containing the target protein. (v) Size exclusion chromatography allows for the separation of EVs on a column depending on size, which may contain a certain amount of free protein etc. [13,30,31]. The advantages and disadvantages of the various separation methods are summarized in Table 1. Today, these methods still inevitably recover soluble impurities in some samples, such as lipoproteins in plasma [32].

## 3. General Function of Small Extracellular Vesicles in Urological Tumors

sEVs are secreted by a variety of different cell types and can play an important role in the development of diseases by transporting proteins, nucleic acids, and lipids. sEVs are involved in cell-to-cell communication, participate in the formation of the tumor microenvironment, affect inflammation and immune regulation, and affect angiogenesis and blood clotting, among numerous other roles [33,34,35,36]. In tumor diseases, sEVs can promote the epithelial-mesenchymal transformation (EMT) process of tumor cells by carrying miRNAs involved in the process [37]. In addition, sEVs plays an important role in tumor metastasis, immune evasion, and tumor resistance mechanisms (Figure 2) [38,39].

### 3.1. sEVs Promotes Angiogenesis in Urological Tumors

Angiogenesis refers to the formation of new blood vessels derived from the original vascular network, a process that is triggered by pro-angiogenic factors [40,41]. Tumor progression is a dynamic process that requires adequate nutrition and oxygen, and angiogenesis is an important mechanism underlying tumor progression [42]. The generation of new blood vessels in the primary tumor lesion can promote the growth and spread of tumor cells. Tumor cells can, in turn, promote angiogenesis by activating endothelial cells. Several studies have shown that miRNAs carried by sEVs can target vascular endothelial growth factor (VEGF), matrix metalloproteinase 2 (MMP2), and MMP9 to regulate angiogenesis [43,44,45,46,47,48]. A previous study by Zhang and coworkers demonstrated that RCC cell-derived sEVs promote the transformation of macrophages to the M2 phenotype, increase the expression of cytokines (such as TGFβ1), enhance the phagocytic ability of macrophages, and induce angiogenesis in RCC by transferring lncARSR, thereby promoting the occurrence and development of RCC [49]. Moreover, αvβ6 integrins in PCa-derived sEVs can be transferred to endothelial cells, which activates TGFβ1 and, results in the inhibition of STAT1 signaling, thereby promoting angiogenesis [50]. Similarly, sEVs secreted by PCa cells have been shown to enhance the angiogenesis and invasion capacity of human umbilical vein endothelial cells, and the same study also suggests that miR-27a-3p may be involved in this phenotypic change [51]. The active cathepsin B protein, which is carried by sEVs can be taken up by endothelial cells by mediating AKT axis phosphorylation, thereby increasing the expression of VEGF and promoting angiogenesis in BC [52]. Li and coworkers found that BC cells in a nutrient-deficient environment secreted sEVs containing glutamine-fructose-6-phosphate aminotransferase 1 (GFAT1) to enhance O-GlcNAcylation in endothelial cells, thus promoting angiogenesis, which indicates a new research direction for developing anti-angiogenic treatments in BC [53]. These studies have jointly shown that angiogenesis promotes tumor growth and infiltration.

### 3.2. sEVs Promotes Epithelial—Mesenchymal Transformation in Urological Tumors

EMT is a process in cancer by which tumor cells acquire invasive and migratory capabilities [54]. EMT primarily manifests by the decreased expression of epithelial markers (such as E-cadherin and β-catenin) and the increased expression of acquired mesenchymal phenotypes (such as vimentin, N-cadherin, and fibronectin) [55,56]. Several studies have shown that tumor-derived sEVs can promote the activation of cancer-associated cells and promote cellular EMT by carrying miRNAs [57,58,59]. A study by Wang and coworkers found that sEVs secreted by CD103+ cancer stem cells could inhibit phosphatase and tensin homolog (PTEN) expression through the delivery of miR-19b-3p, thereby inducing EMT in RCC cells. The migration and invasion ability of RCC tumor cells has also been found to become greatly enhanced after stem cell sEVs treatment [60]. Similarly, sEVs secreted by PC3 cells overexpressing prostate-specific G protein-coupled receptors were shown to promote PCa EMT, thereby promoting migration between PCa cells and normal cells. The exogenous elevation of prostate-specific G protein-coupled receptors (PSGR) occurs in PCa cells, whose secretion of PSGR-carrying sEVs promotes EMT in both PCa and normal prostate epithelial cells [61]. Cancer-associated fibroblasts induce EMT through the secretion of IL-6-containing sEVs, thereby promoting the invasive phenotype of BC [62]. KRT6B expression was found to be elevated in BC-derived sEVs compared to normal tissue-derived sEVs, which promotes EMT in BC, and its high expression results in shorter survival cycles that can be used to predict poor prognosis [63]. Therefore, the results of these studies reveal that EVs can greatly enhance the migration and invasion ability of tumor cells by promoting EMT in urinary tumors, which ultimately promotes the progression of cancer.

### 3.3. Involvement in the Occurrence of Pre-Metastatic Niches in Urological Tumors

Tumor-derived sEVs play a key role in promoting the formation of pre-metastatic niches [64]. Pre-metastatic niches are pro-oncogenic microenvironments that are created by the release of some molecules by the primary tumor including soluble factors, sEVs, and bone marrow-derived cells; these molecules function to regulate the microenvironment, thus making it easier for tumor cells to colonize and spread to distant organs [65,66]. Numerous studies have shown that tumor-derived sEVs, which can alter the function of target cells through the substances they carry, can migrate to target organs via vascular spillover. For example, colorectal cancer (CRC)-derived sEVs are enriched in integrin beta-like 1 (ITGBL1), and when released into circulation, activate fibroblasts to promote the formation of pre-metastatic niches [67,68]. Bone metastasis is an important cause of death in PCa. A study by Wang and coworkers found that the expression of microRNA-378a-3p was significantly elevated in the serum-derived sEVs of patients with bone metastatic PCa, and that miR-378a-3p-containing cells secreted by PC3 cells promoted osteolysis mechanisms via the Dyrk1a/Nfatc1 pathway to enhance bone metastasis in PCa [69]. In addition, it has been shown that tenascin-C is highly expressed in the lymph nodes of patients with BC metastases, and that BC-derived sEVs can induce tenascin-C expression to promote the formation of premetastatic niches [70]. The formation of a premetastatic niche provides many prerequisites for tumor metastasis.

### 3.4. sEVs Regulates the Tumor Microenvironment

The cellular and cell-free components of the tumor microenvironment (TME) can influence tumor development and response to therapy [71]. Key components of the TME include immune cells, stromal cells, blood vessels, and extracellular matrix. sEVs have the effect of promoting inflammatory factor production, tumor angiogenesis, and metastasis in theTME [72]. A study by Yin and coworkers found that colorectal cancer cell-derived sEVs could upregulate PD-L1 in macrophages to promote tumor immune evasion [73]. It has also been shown that sEV-loaded miRNAs can regulate the communication between cancer cells and hepatic stellate cells in the hepatocellular carcinoma tumor microenvironment [74]. Several studies have shown that sEVs can mediate communication between the tumor and the microenvironment via the Notch pathway, for example, in multiple myeloma [75]. Therefore, sEVs play an important role in TME.

### 3.5. Antitumor Effects of sEVs in Urological Tumors

Stem cell-derived sEVs have therapeutic anti-tumor effects. Human liver stem cell-derived sEVs loaded with antitumor miR-145 attenuate the invasive effects of renal stem cells [76]. Similarly, it has been shown that human hepatic stem cell-derived sEVs treated tumor endothelial cells to downregulate the expression of miR-15a, miR-181b, miR-320c, and miR-874, thereby inhibiting the angiogenic effects of tumor endothelial cells [77]. Thus, new prospects for the treatment of urological tumors could focus on stem cell-derived sEVs.

## 4. Application of Small Extracellular Vesicles in the Diagnosis of Urinary Tumors

Compared to traditional tumor markers, sEVs can cross the blood-brain barrier and enter the circulation. This particular feature renders sEVs easily detectable in patients’ biological fluids. sEVs carry an abundance of genetic material, are highly stable and non-degradable in bodily fluids, and have the advantages of having high specificity and readily undergoing extraction. These features have led to the significant adoption of sEV-based biomarkers in the clinical field. The first sEV-based RNA test for prostate cancer has been included in the National Comprehensive Cancer Network guidelines for the early detection of prostate cancer (Table 2) [7,78,79].

### 4.1. miRNAs in Small Extracellular Vesicles

MicroRNAs (miRNAs) are endogenous non-coding RNAs with an average length of 22 nt [107]. They can regulate changes in gene expression by targeting one or more mRNAs, thereby regulating cell growth, coordinating differentiation, and causing functional changes. In sEVs, the content of miRNAs is the highest among all types of RNA. Due to the lipid membrane that envelopes sEVs, miRNAs are protected from degradation by RNA enzymes. Therefore, the miRNA content in sEVs is much higher than that of free miRNA in cells and bodily fluids [108]. These characteristics make miRNAs in sEVs more suitable as biomarkers for tumors than those present in bodily fluids.

There have been numerous studies on sEVs that are obtainable from the urine, serum, and plasma as diagnostic markers of urological tumors. Despite breakthroughs in the advanced treatment of renal cell carcinoma (RCC), the mortality rate of kidney cancer remains high. Currently, there are no reliable biomarkers available for the early diagnosis and prognostic monitoring of RCC. One study that conducted a microarray analysis of sEVs extracted from the serum of patients with advanced RCC showed that miR-4525 contained by sEVs was significantly elevated compared to healthy controls. The investigators hypothesized that miR-4525 in serum-derived sEVs could serve as a potential biomarker for advanced RCC [80]. Meng and coworkers found that sEV extracted from the serum of RCC patients contained higher miR-155 compared to healthy people [81]. In addition to serum-derived sEVs, plasma-derived sEVs also comprise an important source for the diagnosis of RCC. Dias and coworkers found that the two plasma-containing sEV-derived miRNAs hsa-miR-301a-3p and hsa-miR-1293 were expressed at higher levels in patients with metastatic RCC compared to patients with non-metastatic RCC, leading them to propose that the levels of these molecules might serve as biomarkers for metastatic RCC [82]. Similarly, Xiao and coworkers sequenced plasma-derived sEVs from RCC patients and healthy individuals, from which they found that has-miR-92a-1-5p, has-miR-149-3p, and has-miR-424-3p were differentially expressed and of diagnostic value [83]. Urine-derived sEVs also have diagnostic value; Qin and coworkers identified the overexpression of miR-224-5p in urine-derived sEVs from RCC patients, which can be used as a biomarker for immunotherapy [84]. miR-204-5p and miR-30c-5p in urinary sEVs have great potential as biomarkers for the early diagnosis of RCC [85,86].

Prostate cancer (PCa) is the most common male malignancy other than lung cancer. Due to the lack of treatment modalities, advanced PCa is difficult to cure and has a low survival rate compared to early PCa, which has a higher cure rate using surgery combined with hormone therapy [109]. Therefore, the identification of early diagnostic markers for PCa is crucial. sEVs are increasingly being studied as diagnostic markers for tumors. Linuma and coworkers found that miR-93 in serum-derived sEVs was significantly lower in patients after radiotherapy and had a role in monitoring treatment efficacy [87]. Similarly, miR-181a-5p in serum-derived sEVs can be used as a marker of bone metastatic PCa [88]. Plasma-derived sEVs also contain miR-145, miR-221, miR-451a, and miR-141, which have diagnostic potential in PCa [89]. Among them, miR-221 has the ability to sort benign and malignant tumors [90]. For the differential diagnosis of PCa and benign prostatic hyperplasia (BPH), Davey and coworkers identified miR-375 and miR-574 in urine-derived sEVs, the combination of which yield the best diagnostic power out of all the contained miRNAs for the screening of benign and malignant tumors [91]. Similarly, Matsuzaki and coworkers found that miR-30b and miR-126 in urine-derived sEVs predicted PCa with much higher sensitivity and specificity than serum prostate-specific antigen (PSA) [92].

In the diagnosis of bladder cancer (BC), sEVs obtained from blood, urine, and tissue sources are also of diagnostic value. The differential expression of miR-185, miR-106a, and miR-10b in plasma-derived sEVs can predict BC survival [93]. miR-96-5p and miR-183-5p in urinary sEVs can be used in BC diagnosis and follow-up [94].

### 4.2. lncRNA in Small Extracellular Vesicles

Long-stranded non-coding RNAs (lncRNAs) are RNAs longer than 200 nucleotides that do not encode proteins and can be used as tumor markers. Patients with prostate cancer were found to have increased expression of the lncRNA PCA3 in sEVs obtained from their urine; therefore, PCA3 in urine-derived sEVs can be used as a marker for the early diagnosis of PCa [95]. Similarly, there was a significant difference found in the expression of lncRNA-p21 in urine-derived sEVs between BPH and PCa, suggesting that it can be used as a diagnostic marker to distinguish between benign and malignant tumors [96]. The lncRNA HOXD-AS1 in serum-derived sEVs can promote the distant metastasis of PCa, therefore, have predictive value in the metastasis of PCa [97]. The reduced expression of lncRNA PTENP1 in plasma-derived sEVs in patients with bladder cancer may serve as a potential marker for BC [98]. Similarly, the expression of the lncRNAs ANRIL in urine-derived sEVs is significantly elevated in BC patients, and therefore, can be used as a non-invasive diagnostic marker for BC [99].

### 4.3. CircRNAs in Small Extracellular Vesicles

Circular RNAs (circRNAs) are endogenous, non-coding RNAs that are highly conserved and stable. Currently, there have been few studies on circRNAs in sEVs. Xiao and coworkers showed that plasma-derived sEVs harboring circ_400068 were significantly more highly expressed in kidney cancer patients, suggesting that this RNA is associated with kidney cancer progression and has potential as a diagnostic marker [100]. Another study by Li and coworkers found that circ_0044516 expression was upregulated in the sEVs of prostate cancer patients, which could promote PCa proliferation and metastasis. They concluded that this circRNA has diagnostic value for PCa [101].

### 4.4. Proteins in Small Extracellular Vesicles

Tsuruda and coworkers detected the increased expression of RAB27Bin sEVs derived from RCC cells and found it to have a positive correlation with sunitinib resistance, suggesting that this protein could be used as a prognostic marker in RCC [102]. Iliuk and coworkers performed a proteomic analysis of plasma-derived sEVs obtained from RCC patients and found that the phosphorylated form of the protein LYRIC (MTDH) could potentially be used as a biomarker [103]. Polymerase I and transcript release factor (PTRF) expression was found to be higher in the urine-derived sEVs of RCC patients compared to normal human urine-derived sEVs, suggesting that PTRF could be used as a potential diagnostic marker in RCC [104]. Carbonic anhydrase 9 (CA IX) is highly expressed and active in the plasma-derived sEVs of PCa patients compared to normal subjects; therefore, CA IX in sEVs may be a biomarker of PCa progression [105]. It has been shown that heat shock protein 90 (Hsp 90) is significantly upregulated in the urine-derived sEVs of BCa patients and can be used as a diagnostic marker for BCa [106]. In addition, Igami and coworkers found elevated expression of carcinoembryonic antigen-associated adhesion molecule protein (CEACAM) in the urinary sEVs of BCa patients, and that these sEVs could be a new target for liquid biopsy testing [110].

## 5. Investigations of Small Extracellular Vesicles in the Treatment of Urological Tumors

### 5.1. Small Extracellular Vesicles and Tumor Drug Resistance

In clinical practice, one of the major challenges in tumor treatment is tumor drug resistance. Drug-resistant tumors can secrete numerous sEVs that contain resistance-associated proteins, and these sEVs can in turn promote the development of drug resistance. Tinibs (a chemotherapy drug for RCC) are the first-line drugs used for kidney cancer treatment. He and coworkers showed that many advanced RCCs are resistant to sorafenib (a chemotherapy drug for RCC), leading to poor disease treatment outcomes. They also found that sEVs could target MutL homolog 1 (MLH1) by delivering miR-31-5p, thereby leading to sorafenib resistance in RCC. Similarly, they detected the significant upregulation of miR-31-5p in the plasma-derived sEVs of drug-resistant RCC patients [111]. On the other hand, it has been suggested that ketoconazole can inhibit the formation of sEVs in kidney cancer cells as a way to suppress the proliferation and migration function of the tumor. The combination of sunitinib and ketoconazole may improve the therapeutic efficacy of sunitinib [112]. Research by Guang and coworkers showed that miR-423-5p contained in sEVs secreted by cancer-associated fibroblasts targeted GREM2 via the TGFβ pathway, thereby promoting PCa drug resistance [113]. Similarly, sEV-derived miR-27a produced by prostate fibroblasts improved PCa chemoresistance by suppressing P53 gene expression [114]. A study by Shan and coworkers found that cancer-associated fibroblasts secreting sEVs could directly transport miR-148b-3p into bladder cancer cells, thereby promoting BC metastasis, proliferation, and drug resistance [115]. The above studies jointly show that sEV-mediated tumor drug resistance is a phenomenon that can provide numerous new targets for the targeted therapy of urinary tumors, promote the personalized treatment of urinary tumors, and improve treatment efficiency.

### 5.2. Small Extracellular Vesicles as Drug Carriers

Most of the tumor treatment drugs used clinically have the disadvantage that only a small fraction of their dose can reach the lesion to achieve a therapeutic effect. This makes the drug less effective and may cause stronger toxicity and side effects [116]. sEVs can carry a variety of therapeutic substances and easily cross the blood-brain barrier. Numerous studies have shown that macrophage-derived hybrid sEVs can be used to target tumors by carrying relevant antitumor drugs such as Adriamycin [117]. Macrophage-derived sEVs can also serve as a drug delivery system for triple-negative breast cancer by carrying paclitaxel and Adriamycin [118]. Currently, there are no studies on the development of such drug delivery systems in RCC, which still needs to be explored in depth. In contrast, there have been more studies on sEV drug delivery systems in PCa. One study used genetic engineering techniques to design an anti-prostate specific membrane antigen (PMSA) sEV that could target late-stage PCa to organize cellular internalization [119]. Similarly, Wang and coworkers used genetic engineering techniques to reverse the tumor microenvironment of PCa by encapsulating the sonosensitizers Chlorin e6 and the immune adjuvant R848 into sEV [120]. Zhou and coworkers designed a drug delivery system for macrophage-derived sEVs harboring CD73 inhibitors and monoclonal antibodies target to programmed cell death ligand 1. The combination of this complex significantly inhibited the activation and infiltration of cytotoxic T lymphocytes in BC [121]. Pelvic radiotherapy is an important treatment modality for prostate cancer, where acute radiation cystitis is a common response to radiotherapy. It has been shown that mesenchymal stem cells (MSCs) can target fibrosis, inflammation, and angiogenesis in cystitis to achieve a therapeutic effect [122]. Zhao and coworkers induced that nano-sEVs released from MSCs standardized for pluripotent stem cells could be used to treat prostate cancer [123]. Today, in addition to the therapeutic benefits of stem cell and immune cell-derived sEVs, milk-derived sEVs also have corresponding therapeutic benefits. For example, milk-derived sEVs are stable, can be absorbed by the intestine, and remain intact and improve the intestinal barrier [124]. In addition, sEVs from sources such as goat and donkey milk also have anti-inflammatory and immunomodulatory abilities and can therefore be used extensively for the regulation of chronic diseases [125]. The above studies clearly indicate that sEVs have not yet been well studied in the treatment of urological tumors, but their unique physiological properties make them very promising for such research. On the other hand, food-derived sEVs can enhance cell targeting by mild modification, which is one of the prospects for oral treatment of tumors (Figure 2).

### 5.3. Small Extracellular Vesicles and Tumor Vaccines

Vaccines developed using cancer-associated cell-derived sEVs have higher affinities than conventional vaccines. Numerous studies have shown that dendritic cell (DC)-derived sEVs can be used as effective anti-tumor vaccines. sEVs have previously been designed to function as an in-situ DC-initiated vaccine to boost anti-tumor immunity in breast cancer [126]. Similarly, Lu and coworkers demonstrated that antigen-modified DC-derived sEVs could inhibit tumor regression in hepatocellular carcinoma [127]. In contrast, there have been fewer studies conducted on urological tumors. Xu and coworkers found that Reca cell-derived sEVs could stimulate CD8+ T cells to enhance the anti-renal cortical adenocarcinoma effect via the Fas ligand (FasL) signaling pathway [128]. In addition, oral particulate vaccines encapsulating tumor-associated antigens derived from mouse prostate cancer cell lines were combined with cyclophosphamide to significantly reduce the tumor volume of PCa [129]. PCa-derived sEVs modified with interferon-γ into a tumor vaccine were shown to increase the number of M1 macrophages and thus significantly inhibit tumor growth [130]. Currently, there is insufficient research on tumor vaccines, which comprise a highly promising new approach to tumor treatment. It is not difficult to hypothesize that sEVs loaded with tumor suppressor genes and tumor chemotherapy drugs will contribute greatly to the development of such tumor vaccines.

## 6. Conclusions

The early diagnosis of urological tumors is one of the key factors in improving patient survival and prognosis. sEVs are novel liquid biopsy markers for urological tumors. They are present in a variety of bodily fluids and tissues and are highly heterogeneous. Therefore, sEVs can potentially be used as biomarkers for non-invasive screening. Because of their phospholipid bilayer, which transports and protects various bioactive substances within, and ease in crossing the blood-brain barrier, sEVs have become a focus of research in the development, diagnosis, and treatment of diseases. Tumor cell-derived sEVs regulate angiogenesis, epithelial-mesenchymal transition, and the microenvironment of urological tumors by carrying substances such as nucleic acids and proteins. An increasing number of studies have focused on sEVs in the diagnosis of urological tumors. sEVs also contribute to EMT, angiogenesis, and drug resistance in urological tumors. Such studies related to sEVs provide new ideas for the diagnosis and treatment of urological tumors and hold great promise for further research. Research on sEV in the diagnosis of urological tumors has become more extensive and focused over time, but most of the current research remains only at the laboratory stage, rather than the clinical stage. This is mainly due to the lack of a definitive molecule that has been repeatedly confirmed to be useful across different studies, as well as the lack of clinical data. Some studies have focused on the therapeutic resistance of sEVs in urological tumors. However, there is a paucity of studies on the application of sEVs in the treatment of urological tumors, particularly in renal cancer, concurrent with a lack of data on translational investigations of their clinical application. Therefore, further studies are needed to investigate sEV-mediated tumor vaccines and tumor drug carriers, as well as to understand the impact and mechanisms of sEV-mediated tumor resistance on targeted therapy in urological tumors.

## Figures and Tables

**Figure 1 cancers-15-00100-f001:**
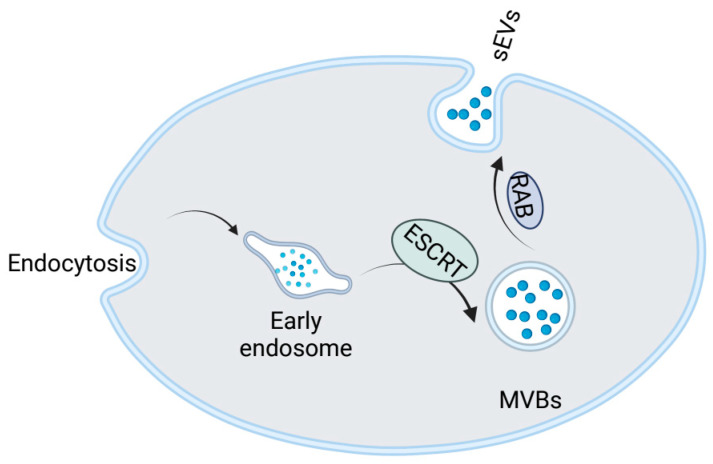
Mechanism of sEVs formation. This figure summarizes the formation process of sEVs. sEVs form MVBs via the endosomal pathway, which are then released into the extracellular space by binding MVBs to Rab 27a and 27b. (This figure was created with Biorender.com).

**Figure 2 cancers-15-00100-f002:**
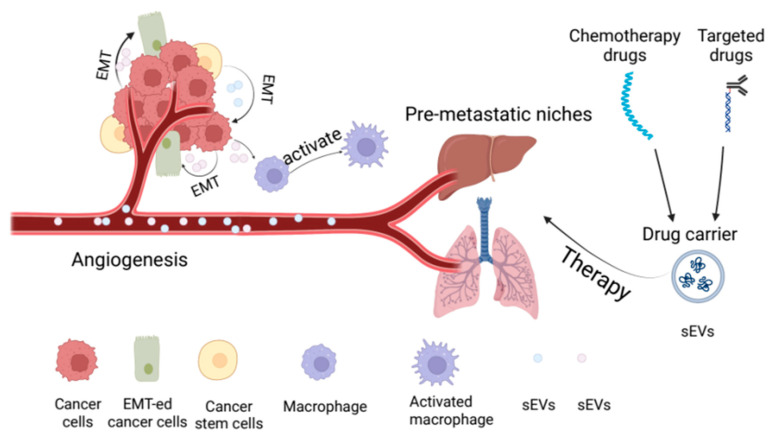
The general function of sEV in tumors and its use in therapy. This figure summarizes the role of sEVs in urological tumors and their application in the treatment. Urological tumor cells and sEVs released by urological cancer stem cells can promote tumor cell EMT, endothelial cell angiogenesis, macrophage activation, and pre-metastatic niches formation. sEVs can be used as drug delivery systems to treat urological tumors. (This figure was created with Biorender.com).

**Table 1 cancers-15-00100-t001:** Separation methods for sEVs.

Method	Mechanism	Advantages	Disadvantages
Ultracentrifugation	Density	Gold standard; Low cost	Time consuming; low specificity
Density gradients	Density	Gold standard; High specificity	Low production; Time consuming
Precipitation	Solubility	Quikly	Low specificity, presence of protein
Immuno-capture	Antigen	Quikly	High cost; High specificity
Size exclusion chromatography	Size	Quikly	Contaminated protein

**Table 2 cancers-15-00100-t002:** Possible candidate markers of EVs in Urological tumors.

Type	Disease	Source	Cargoes	Reference
miRNA	RCC	Serum	miR-4525, miR-155	Muramatsu-Maekawa et al., Meng et al. [80,81]
		Plasma	miR-301a-3p, miR-1293, miR-92a-1-5p, miR-149-3p, miR-424-3p	Dias et al., Xiao et al. [82,83]
		Urine	miR-224-5p, miR-204-5p, miR-30c-5p	Qin et al., Kurahashi et al., Song et al. [84,85,86]
	PCa	Serum	miR-93, miR-181a-5p	Iinuma et al., Wang et al. [87,88]
		Plasma	miR-145, miR-221, mIR-451a, miR-141	Zabegina et al., Kim et al. [89,90]
		Urine	miR-375, miR-574, miR-30b, miR-126	Davey et al., Matsuzaki et al. [91,92]
	BC	Plasma	miR-185, miR-106a, miR-10b	Sabo et al. [93]
		Urine	miR-96-5p, miR-183-5p	El-Shal et al. [94]
lncRNA	PCa	Urine	lncRNA PCA3, lncRNA-p21	Li et al., Işın et al. [95,96]
		Serum	lncRNA HOXD-AS1	Jiang et al. [97]
	BC	Plasma	lncRNA PTENP1	Zheng et al. [98]
		Urine	lncRNA ANRIL	Abbastabar et al. [99]
CircRNA	RCC	Plasma	circ_400068	Xiao et al. [100]
	PCa	Blood	circ_0044516	Li et al. [101]
Protein	RCC	Cells	Rab 27b	Tsuruda et al. [102]
		Plasma	MTDH	Iliuk et al. [103]
		Urine	PTRF	Zhao et al. [104]
	PCa	Plasma	CA IX	Logozzi et al. [105]
	BCa	Urine	Hsp 90, CEACAM	Tomiyama et al. [106]

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
