# Peer review of "Current Status of Research on Small Extracellular Vesicles for the Diagnosis and Treatment of Urological Tumors"

_cancers, 2022, doi:10.3390/cancers15010100_

Round 1

Reviewer 1 Report

This is an interesting review on the status of research on small extracellular vesicles, with particular reference to urological tumours. However, there are some limitation:

-          In the “Overview paragraph” Authors should better explained what they meant with the sentence: “sEV occupies a major position”

-          Authors should include add more figures and tables in order to make the paper more interesting and appealing for readers.

-          References should be updated.

-          In the “Promotion of angiogenesis paragraph” Authors should update the literature with the most recent papers on the topic and comment them (such as Prigol AN, Rode MP, Silva AH, Cisilotto J, Creczynski-Pasa TB. Pro-angiogenic effect of PC-3 exosomes in endothelial cells in vitro. Cell Signal. 2021 Nov;87:110126. doi:10.1016/j.cellsig.2021.110126. Epub 2021; Li X, Peng X, Zhang C, Bai X, Li Y, Chen G, Guo H, He W, Zhou X, Gou X. Bladder Cancer-Derived Small Extracellular Vesicles Promote Tumor Angiogenesis by Inducing HBP-Related Metabolic Reprogramming and SerRS O-GlcNAcylation in Endothelial Cells. Adv Sci (Weinh). 2022 Oct;9(30):e2202993. doi: 10.1002/advs.202202993. Epub 2022 Aug 31. PMID: 36045101; PMCID: PMC9596856.Aug 30. PMID: 34474113).

-          In the  “Promotion of epithelial mesenchymal transition” paragraph Authors should update the literature with the most recent papers on the topic and comment them (such as Song Q, Yu H, Cheng Y, Han J, Li K, Zhuang J, Lv Q, Yang X, Yang H. Bladder cancer-derived exosomal KRT6B promotes invasion and metastasis by inducing EMT and regulating the immune microenvironment. J Transl Med. 2022 Jul 6;20(1):308. doi: 10.1186/s12967-022-03508-2. PMID: 35794606; PMCID: PMC9258227. Li Y, Li Q, Li D, Gu J, Qian D, Qin X, Chen Y. Exosome carrying PSGR promotes stemness and epithelial-mesenchymal transition of low aggressive prostate cancer cells. Life Sci. 2021 Jan 1;264:118638. doi: 10.1016/j.lfs.2020.118638. Epub 2020 Oct 24. PMID: 33164833)

-          In the  “Involvement in the occurrence of pre-metastatic niches” paragraph Authors should update the literature with the most recent papers on the topic and comment them (such as  Wang J, Du X, Wang X, Xiao H, Jing N, Xue W, Dong B, Gao WQ, Fang YX. Tumor-derived miR-378a-3p-containing extracellular vesicles promote osteolysis by activating the Dyrk1a/Nfatc1/Angptl2 axis for bone metastasis. Cancer Lett. 2022 Feb 1;526:76-90. doi: 10.1016/j.canlet.2021.11.017. Epub 2021 Nov 19. PMID: 34801597. Silvers CR, Messing EM, Miyamoto H, Lee YF. Tenascin-C expression in the lymph node pre-metastatic niche in muscle-invasive bladder cancer. Br J Cancer. 2021 Nov;125(10):1399-1407. doi: 10.1038/s41416-021-01554-z. Epub 2021 Sep 25. PMID: 34564696; PMCID: PMC8575937) 

-          The discussion section should be improved taking into account also the new references added.

-         

Author Response

Response to reviewer1 comments

Point 1: In the “Overview paragraph” Authors should better explained what they meant with the sentence: “sEV occupies a major position”

Response 1: The dominance of sEVs is mainly due to the thorough study of the mechanism of sEVs, and there are many related studies. Therefore, we change the text in paragraph 2 to read “Of these three types, research has primarily focused on sEVs, and this is mainly because their formation pathways have been more thoroughly studied and they are more stable in bodily fluids. Moreover, sEVs can act asbridges for cellular communication between donor and recipient cells”.

Point 2: Authors should include add more figures and tables in order to make the paper more interesting and appealing for readers.

Response 2: We added a table at the end of the article. See annex for details.

Point 3: References should be updated.

Response 3: We added the references requested by reviewer 1 to paragraphs 3.1, 3.2, and 3.3, specifically cited as references 51, 53, 63, 61, 69, and 70. See annex for details.

Point 4: The discussion section should be improved taking into account also the new references added.

Response 4: Minor changes have been made to the concluding paragraphs. The full text has been professionally polished.

Reviewer 2 Report

The review by Zhang and coworkers describes the role of small extracellular vesicles (sEVs) in the context of urological tumors. The authors explore the contribution of sEVs to different biological processes related to tumorigenesis, such as angiogenesis, epithelial-mesenchymal transition, and formation of pre-metastatic niches. Moreover, they report the significance of sEVs for both the diagnosis, as carriers of biomarkers, and therapy of urological tumors.

According to me, the first part of the review is too general and not so accurate and informative, whereas the description of the role of sEVs in the specific context of urological tumors is more interesting. Also, the title is too general and vague. I would recommend focusing more on the specific context of the urological tumors, giving more emphasis on this topic, and changing the title accordingly as well as improving the description and/or discussion of the key aspects listed below. 

I also recommend English language editing and proofreading, and careful revision of the use of abbreviations throughout the text.

Major points:

1)     In my opinion the background is very short and general and needs extensive revision. For example, I would introduce here the state of the art regarding the diagnosis and therapy of urological tumors, apart from EVs (free biomarkers, clinical trials), which is reported in different parts of paragraphs 4 and 5, but it could be highlighted and summarized here.

Line 46-53: the EV description is really general. Please introduce better the topic.

2)     Paragraph 2 is too general and not so accurate and needs careful revision.  I would suggest revising the title (consider also substituting the term overview), and introducing inside the title “urological tumors”. I would also revise the division in subparagraphs, which are a bit too short. Maybe subparagraph 2.3 could be fused directly with paragraph 3.

lines 60-66 (and also Lines 22-23 of the Abstract): these characteristics are not specific to sEVs, but belong to all types of EVs. Please correct.

lines 75-84: the term sEV refers to the size and not the origin of EVs. sEVs can be formed by both the endocytic pathway and the membrane blebbing followed by vesicle budding. This other pathway should be also mentioned. It is true, instead, that microvesicles that are larger than sEVs are mostly released from the membrane and not through the endocytic pathway, due to physical constraints. See also MISEV 2018 on EV classification.

I appreciate the topic introduced in Paragraph 2.4, that is very relevant, but it needs more discussion on the isolation techniques and also on the choice of samples, i.e urine or blood, to have the best results in terms of EV purification, amount and significance in the context of urological tumors.

3)     As for paragraph 2, paragraph 3 is a bit too general. I would suggest revising it, mentioning in the text urological tumors, changing the paragraph subdivision, and adding a part on the description of the antitumoral effect that sEVs can have. 

4)     Are some of the potential biomarkers listed in clinical trials?

5)     Paragraph 5.2 is a bit repetitive at the beginning, please revise. Moreover, here it is important to discuss the different typologies of sEVs of different origins that could be used for drug delivery. In the beginning, MSC-derived Evs were principally used, but now other sources, including milk or green renewable sources, such as plants or microalgae have been widely investigated. Please discuss this important aspect.

6)     Figures:

Figure 1 reports the endocytic pathway which refers to exosomes and not sEVs, that is an operational term that refers just to size and not to origin, as already highlighted above (see MISEV2018).

I could not find Figure 2 mentioned in the text. Please check. Moreover, I would focus Figure 2 specifically on urological tumors. Please consider also that EVs, i.e. the ones released by immune cells, can also have an antitumoral effect. sEVs are not only related to the development of pathology and revise the Figure and/or the title accordingly.

Minor points:

1) Please specify always abbreviations the first time they are used in the text (i.e. MVB at line 68, PTEN at line PC3 at line 136).

2) After introducing an abbreviation, please be sure of keeping using it (i.e. small extracellular vesicles at lines 86, 96, 103, and in many other cases are still used instead of sEVs).

3) In many cases EVs (plural) should be used instead of EV.

4) Check that the reference number in the text is before and not after the dot. 

Line 67: I would specify “EV proteins”.

Line 69: I would specify “EV lipids”.

Lines 72-73: this sentence is too vague, without a reference, and partial. sEVs are also released by normal cells and contribute to physiological communication among cells. I would delete the sentence or at least rephrase it.

Line 104: tumor or tumour? Please choose

Line115: I would suggest using “and coworkers” instead of “et al.” in the text.

Line 229: I would remove stranded

Lines 272-274: the sentence is vague and I would suggest supporting it with references showing the general implication of sEVs in drug resistance, not focused on urinary tumors.

Line 275: please explain what are Tinibs and Sorafenib.

Line 336: I would suggest substituting “Discussion” with “Conclusions”.

Author Response

Response to reviewer1 comments

Point 1: In my opinion the background is very short and general and needs extensive revision. For example, I would introduce here the state of the art regarding the diagnosis and therapy of urological tumors, apart from EVs (free biomarkers, clinical trials), which is reported in different parts of paragraphs 4 and 5, but it could be highlighted and summarized here.

Line 46-53: the EV description is really general. Please introduce better the topic.

Response 1: The content of paragraphs 4 and 5 is summarized in the background paragraph, which is detailed in the annex.

Point 2:  Paragraph 2 is too general and not so accurate and needs careful revision.  I would suggest revising the title (consider also substituting the term overview), and introducing inside the title “urological tumors”. I would also revise the division in subparagraphs, which are a bit too short. Maybe subparagraph 2.3 could be fused directly with paragraph 3.

lines 60-66 (and also Lines 22-23 of the Abstract): these characteristics are not specific to sEVs, but belong to all types of EVs. Please correct.

lines 75-84: the term sEV refers to the size and not the origin of EVs. sEVs can be formed by both the endocytic pathway and the membrane blebbing followed by vesicle budding. This other pathway should be also mentioned. It is true, instead, that microvesicles that are larger than sEVs are mostly released from the membrane and not through the endocytic pathway, due to physical constraints. See also MISEV 2018 on EV classification.

I appreciate the topic introduced in Paragraph 2.4, that is very relevant, but it needs more discussion on the isolation techniques and also on the choice of samples, i.e urine or blood, to have the best results in terms of EV purification, amount and significance in the context of urological tumors.

Response 2: Paragraph 2.3 had been deleted and the content had been merged into paragraph 3. See the attached document for details. The classification of EVs has been revised and sEVs have been defined as exosomes that are not specifically referred to in the paper. See paragraph 2, fifth line. Therefore, Figure 1 does refer to the endocytosis pathway of exosomegenesis. A discussion of the advantages and disadvantages of separation techniques was added, as shown in paragraph 2.3, and a summary was provided in Table 1, as shown in the Annex. The term urological tumor was added to the title, see 3 throughout the title section.

Point 3:  As for paragraph 2, paragraph 3 is a bit too general. I would suggest revising it, mentioning in the text urological tumors, changing the paragraph subdivision, and adding a part on the description of the antitumoral effect that sEVs can have.

Response 3: The addition of paragraph 3.5 is detailed in the annex.

Point 4: Are some of the potential biomarkers listed in clinical trials?

Response 4: All potential clinical markers have been listed here.

Point 5: Paragraph 5.2 is a bit repetitive at the beginning, please revise. Moreover, here it is important to discuss the different typologies of sEVs of different origins that could be used for drug delivery. In the beginning, MSC-derived Evs were principally used, but now other sources, including milk or green renewable sources, such as plants or microalgae have been widely investigated. Please discuss this important aspect.

Response 5: Paragraph 5.2 removes some of the repetition at the beginning and adds therapeutic effects of MSC-derived sEVs and milk-derived sEVs, as detailed in the Annex.

Point 6: Figures:

Figure 1 reports the endocytic pathway which refers to exosomes and not sEVs, that is an operational term that refers just to size and not to origin, as already highlighted above (see MISEV2018).

I could not find Figure 2 mentioned in the text. Please check. Moreover, I would focus Figure 2 specifically on urological tumors. Please consider also that EVs, i.e. the ones released by immune cells, can also have an antitumoral effect. sEVs are not only related to the development of pathology and revise the Figure and/or the title accordingly.

Response 6: The answer in Figure 1 is shown in response 2. Figure 2 has already marked the parts mentioned in the original article.

Point 7: 1) Please specify always abbreviations the first time they are used in the text (i.e. MVB at line 68, PTEN at line PC3 at line 136).

2) After introducing an abbreviation, please be sure of keeping using it (i.e. small extracellular vesicles at lines 86, 96, 103, and in many other cases are still used instead of sEVs).

3) In many cases EVs (plural) should be used instead of EV.

4) Check that the reference number in the text is before and not after the dot.

Line 67: I would specify “EV proteins”

Line 69: I would specify “EV lipids”.

Lines 72-73: this sentence is too vague, without a reference, and partial. sEVs are also released by normal cells and contribute to physiological communication among cells. I would delete the sentence or at least rephrase it.

Line 104: tumor or tumour? Please choose

Line115: I would suggest using “and coworkers” instead of “et al.” in the text.

Line 229: I would remove stranded

Lines 272-274: the sentence is vague and I would suggest supporting it with references showing the general implication of sEVs in drug resistance, not focused on urinary tumors.

Line 275: please explain what are Tinibs and Sorafenib.

Line 336: I would suggest substituting “Discussion” with “Conclusions”.

Response 7: The full name of the specified abbreviation has been supplemented. Reference numbers are placed before dots. Unified use of sEVs and tumors. All abbreviations are used in normal. Discussion was changed to Conclusion. Tinibs and Sorafenib refer to chemotherapy drugs for RCC, as indicated in the text. We use EVs instead of EV. The whole text is professionally polished.

Round 2

Reviewer 2 Report

The article strongly improved and is now more focused, precise, and informative. Some points are still open:

- In my opinion, the title is too broad. I would directly focus on urological tumors. i.e. “Current status of research on small extracellular vesicles for the diagnosis and treatment of urological tumors”.

- The following sentence is not really clear to me” The sEVs that appear in this article specifically refer to exosomes that are not specifically described [12-14]” (Lines 68-70). Moreover, it seems to break the links between the previous and the next sentence. I suggest removing it and simply using throughout the text the term sEVs, which is more appropriate than exosomes, as explained in the MISEV 2018 that authors do cite.  

-   As discussed in my previous revision, sEVs can also form through budding from the membrane. I would add a sentence on this at the end of paragraph 2.2.

- Finally, the paper still needs a bit more English language editing and proofreading:

-       the term “sEVs”, which is plural, is often followed by third person singular verb (i.e. lines 126 and 153). Please revise it

- Please revise even more carefully the use of abbreviations

i.e. line 54: substitute prostate cancer with PC already used in line 39.

line 154: EMT abbreviation has been already introduced in line 123.

-       Please revise spacing (lines 72, 155, 160……etc)

-       I would suggest using “and coworkers” instead of “et al.” in the text. (see line 135).

-       Line 219: eliminate the comma

Author Response

Response to reviewer2 comments

Point 1: In my opinion, the title is too broad. I would directly focus on urological tumors. i.e. “Current status of research on small extracellular vesicles for the diagnosis and treatment of urological tumors”.

Response 1: We have changed the title to " Current status of research on small extracellular vesicles for the diagnosis and treatment of urological tumors ".

Point 2: The following sentence is not really clear to me” The sEVs that appear in this article specifically refer to exosomes that are not specifically described [12-14]” (Lines 68-70). Moreover, it seems to break the links between the previous and the next sentence. I suggest removing it and simply using throughout the text the term sEVs, which is more appropriate than exosomes, as explained in the MISEV 2018 that authors do cite.  Response 2: We have removed this sentence.

Point 3: As discussed in my previous revision, sEVs can also form through budding from the membrane. I would add a sentence on this at the end of paragraph 2.2.

Response 3: We add a sentence on this at the end of paragraph 2.2.

Point 4: the term “sEVs”, which is plural, is often followed by third person singular verb (i.e. lines 126 and 153). Please revise it

Response 4: We have modified these. 

 Point 5: Please revise even more carefully the use of abbreviations

i.e. line 54: substitute prostate cancer with PC already used in line 39.

line 154: EMT abbreviation has been already introduced in line 123.

Response 5: We have modified these.

Point 6: Please revise spacing (lines 72, 155, 160……etc)

Response 6: We have modified these.

Point 7: I would suggest using “and coworkers” instead of “et al.” in the text. (see line 135).

Response 7: We have modified these.

Point 8: Line 219: eliminate the comma

Response 7: We have modified these.
